# Comparison of Effects of Implicit versus Explicit Learning of a Novel Skill in Young Gymnastic Athletes

**DOI:** 10.3390/bs14090798

**Published:** 2024-09-10

**Authors:** Henrik Borge Garnaas, Roland van den Tillaar

**Affiliations:** Department of Sports Sciences and Physical Education, Nord University, 7600 Levanger, Norway; henrikbg@hotmail.com

**Keywords:** dual task, gymnastics, front flip, retention test

## Abstract

Background: The purpose of this study was to compare the effects of explicit learning with implicit learning using a dual-task paradigm on learning a novel skill and whether the performance was maintained over an extended period. Methods: Forty-four young children from a local gymnastics club (*n* = 44, boys *n* = 10, girls *n* = 34, age: 10 ± 2.9 years) completed four 30 min front-flip practice sessions over four weeks between pre- and post-test, followed by two retention tests three and six months after the post-test, in which no front-flip practice occurred. Results: Comparable improvements were found beyond baseline performance for both learning conditions over the six-month hiatus. While both groups increased performance at the post-test, neither explicit nor implicit learning groups were able to maintain the achieved performance level into six months of retention. In particular, the explicit group showed a more marked decrease than the implicit group after three months, which was probably caused by the decay of their reliance on the retrieval of declarative knowledge from working memory. Conclusions: The current findings highlight the importance of conducting retention tests over an extended period to monitor performance development after the acquisition of a novel task by different learning methods, as they can differ over time.

## 1. Introduction

There are several different ways to learn motor skills according to the literature. Explicit learning involves learning through conscious awareness and declarative knowledge of motor tasks [1]. Coaches often use this method to quickly provide information and facilitate rapid motor control improvements [2]. However, explicit learning relies heavily on cognitive resources like working memory due to the use of verbal feedback and instructions [3].

In contrast, implicit learning does not require explicit rules of execution and does not involve declarative knowledge [4]. Riding a bicycle is a classic example of implicit learning, where one performs the skill without conscious attention to the specific movements. Implicit learning results in the accumulation of procedural knowledge, where performance occurs automatically and independently of working memory [5]. To prevent the learner from using working memory to gain declarative knowledge during the learning process in implicit learning, Masters [6] proposed a dual-task paradigm for the implicit learning method. In this approach, the learner is distracted by a concurrent cognitive task during practice, preventing conscious control over their movement. This contributes to the performer being unaware of errors and ignoring task-irrelevant cues rather than relying on correcting feedback and instructions retrieved from working memory. Studies that investigated this were mainly carried out in golf putting, table tennis, throwing, kicking, jumping, grasping and balancing [7,8,9]. Implicit learning has shown advantages compared with explicit learning, such as being less prone to interference from psychological stress [6,7,10], independent of IQ [1], convergent with explicit learning over time [10], stable under fatigue [11] and providing experiences of successful movement performance [12]. This approach is increasingly applied in sports [7].

Collectively, these results reveal several advantages of learning a motor skill implicitly. However, in most implicit vs. explicit studies, no retention tests were performed to examine if learning was stable after a period of time, which is a part of the definition of learning [13]. Furthermore, Maxwell, Masters and Eves [10] did not demonstrate a significant difference between implicit and explicit learners, with a retention test after just three days, while Poolton, Masters and Maxwell [8] showed a retention of performance at a comparable level between implicit and explicit learning methods after a one-year hiatus. The first study had a very short retention period in which the explicit learning group could still use their learned declarative knowledge and thereby have the same results as the implicit learners, while after a one-year hiatus, performance between the two groups could be diminished due to the long period of absence. Nothing is known of a retention period between these two extremes.

Hence, this study aims to compare the effects of explicit learning and implicit learning using a dual-task paradigm on learning a novel skill in gymnastics in young gymnasts. With additional retention tests to assess how performance is maintained over time. It was hypothesised that explicit learners will increase their performance more after the training period than implicit learners due to the possible benefit from feedback and instructions during practice, while implicit learners will encode all actions regardless of their potential outcome without relying on working memory [7]. However, concerning retention, a decline in both groups is expected and that longer retention intervals will result in greater declines in performance levels. Thus, implicit learners will maintain their front-flip performance better as they accumulate procedural knowledge directly, while explicit learners may rely on declarative knowledge stored in working memory.

## 2. Materials and Methods

### 2.1. Participants

This study involved forty-four children (boys *n* = 10, girls *n* = 34) from different groups within a local gymnastics club (age: 10 ± 2.85 years) who participated based on their attendance in gymnastics as a leisure time activity. Of these, 28 participants (aged 8 ± 1.76 years) trained for one hour weekly, and 16 participants (age: 11 ± 1.43) regularly trained for three hours a week. Although the participants had some experience with the front flip, they had never been provided with a structured training program before this experiment. Informed consent was obtained prior to testing from all participants and parents, conforming to the latest revision of the Declaration of Helsinki with approval of the Norwegian Centre for Research Data (project nr: 397032). The consent informed that participating in this project meant that the participants were not allowed to practice a front flip outside the training situation in any form as long as the experiment lasted.

### 2.2. Procedure

The experimental design included pre- and post-tests of a four-week intervention, followed by two retention tests, three and six months after the post-test in which the task was not practised. A standardised warm-up procedure was used at all testing sessions, followed by three attempts to perform a front flip per test per participant, where the best attempt was used for further analysis. To induce some pressure (psychological stress) during the test, it took place in the presence of the other gymnastic athletes, who served as an audience. This psychological stress was added, as, in many situations, children are put under pressure when showing their skills, especially in gymnastics in competition. It has been shown by Masters [6] that under-stress performance improved in the implicit group, whereas the performance of the explicit group degraded. Therefore, it was ecologically valuable to include the presence of other gymnastic athletes to create some pressure. Since the participants may have developed some procedural knowledge from different background experiences, a baseline test before the pre-test was conducted to obtain an indication of the participants’ performance level. All attempts from the participants were in the same order at all tests and filmed using a standard digital camera (Sony Cybershot DSC-W380, Tokyo, Japan) placed on a tripod three metres to the side of the mini-tramp (Figure 1). The performance on video was then assessed by two judges (experienced judges in gymnastics) who were blinded to what group each candidate belonged to. The assessment of the front-flip performance execution was based on the International Gymnastics Federation scoring system known as the ‘code of points’. However, our own code was designed for this experiment. The value of a perfectly executed front flip was placed at 15 points in total, and deductions were related to errors in height, mid-air form and landing categories. Errors were judged to be none, small, medium or large, and, respectively, 0-, 1-, 3- and 5-point deductions in each category were applied. The total score based upon sum of the three categories for each participant was used for analysis

### 2.3. Training Procedure

After the pre-test, based on rank upon the performance of the pre-test, a stratified randomised selection was performed, which ensured sufficient group comparability in which the group was equally divided into an explicit training group (*n* = 22: 5 boys and 17 girls) and an implicit training group (*n* = 22: 5 boys and 17 girls). Each group underwent four training sessions of 30 min each in four weeks before the post-test session. The groups were separated from each other during practice, as the implicit training group were supposed to learn the skill without picking up the underlying rules of performance from the explicit training group. While one group was practising the front flip, the other was practising on uneven and high bars in another hall. The task was constrained to enhance skill acquisition for both groups during practice (Figure 1).

First, a mat was placed in front of the mini-tramp to allow the participants to perform a long jump onto the mini-tramp. Second, a box was placed between the mini-tramp and the jump mat to make them jump higher. Third, hula hoops were placed with a certain space between them. The distance in the gap increased as the participant approached the mini-tramp. The purpose was to get the participant to run faster as he or she approached the mini-tramp. Fourth, jump ropes in three different colours to mirror traffic lights (red, yellow and green) were placed alongside the approach run to signal the increase in running speed from slow, faster to a fast speed phase. During the tests, this equipment was not included to avoid extra constraints. The explicit group received after each trial standardised feedback presented through a set of specific instructions on how to do a front flip: ‘Run faster, as you approach the mini tramp’, ‘have a long jump into the mini tramp’, ‘jump high’, ‘grab your knees’, ‘spin on the apex of the jump’ and ‘open up before landing’. The implicit group did not receive any feedback. At the start of each attempt (start running), the implicit group did move in a random order a relatively simple mathematical additive solving, below 100, like 31 + 19 and 20 + 30, by the mini-tramp safety instructor, which they had to answer when they had landed again.

### 2.4. Statistical Analysis

The normality of data distribution was examined and confirmed using the Shapiro–Wilk test. To assess the effects of the learning methods, a 2 (method: implicit and explicit) × 4 (tests: repeated measures) analysis of variance (ANOVA) was used. In addition, to investigate the development per method, a one-way ANOVA with repeated measures (test occasion) was performed. Where the sphericity assumption was violated, the Greenhouse–Geisser adjustments of the *p*-values were reported. Post hoc testing using Holm–Bonferroni probability adjustment was used to locate significant differences. To investigate the scores between the judges, a one-way ANOVA with repeated measures was used to investigate an eventual systematic bias between the judges together with the limits of agreement, which was calculated using an intraclass correlation coefficient (ICC). If the ICC was <0.5, it indicated poor reliability. Between 0.5 and 0.75 indicated moderate reliability, between 0.75 and 0.9 indicated good reliability, and any value above 0.9 indicated excellent reliability [14]. The effect size was evaluated with Eta partial squared where 0.01 < η_p_^2^ < 0.06 constituted a small effect, a medium effect when 0.06 < η_p_^2^ < 0.14 and a large effect when η_p_^2^ > 0.14 [15]. The level of significance was set at *p* ≤ 0.05. Statistical analysis was performed using SPSS, version 27.0 (IBM Corporation, Armonk, NY, USA).

## 3. Results

The overall difference in points between judges was statistically significant (F = 192.2, *p* = 0.001 *, η_p_^2^ = 0.817), with a mean difference of 0.97 points. However, the ICC was 0.953 (Figure 2).

Due to the high internal consistency between the judges, the average score from the two judges was used for further analysis. Significant effects of test occasion (F = 119, *p* ≤ 0.001, η_p_^2^ = 0.74) and interaction effect (F = 8.87, *p* ≤ 0.001, η_p_^2^ = 0.17) were found, with no significant effect between the two groups (F = 0.7, *p* = 0.414, η_p_^2^ = 0.02). Post hoc comparisons revealed increased performance from the pre-test to the post-test in both groups. Performance declined for both groups from the post-test to the first retention test; however, the decrease was significantly larger in the explicit training group compared with the implicit group (*p* < 0.001). From the first to the second retention test, the implicit training group significantly continued to decrease front-flip performance (*p* < 0.001). However, the total decrease in performance of the explicit training group from the post-test to the test six months later was still significantly more than the total decrease in the implicit training group (2.1 vs. 1.0, *p* = 0.0015, Figure 3).

## 4. Discussion

This study aimed to compare the effects of explicit learning with implicit learning using a dual-task paradigm in the acquisition of a novel gymnastic skill among young gymnastic athletes and assess if the performance was maintained over an extended period. The main findings were that both groups increased front-flip performance after four weeks of intervention. However, in the long term, both groups experienced a decline in performance, with the explicit learning group showing a more noticeable decrease, particularly during the first three months (Figure 3). Despite this decline, both groups still demonstrated a better front-flip performance level after six months than before the intervention. These findings align with previous studies that have investigated explicit and implicit learning through a dual-task paradigm [7].

As both training groups improved their performance from the baseline performance level, this indicates that the number of trials was sufficient for young gymnasts to build a sufficiently large pool of positive action outcomes, leading to the consolidation of procedural knowledge regardless of being learned as implicit or explicit learners. This is particularly significant for implicit learners; as Berry and Broadbent [16] have noted, they must encode all action outcomes. Explicit learners may have benefited from error correction, which helps with the conscious selection of positive action outcomes and avoidance of negative ones.

However, the significant decrease in performance observed for both groups after three months (Figure 3) suggests an insufficient consolidation of procedural knowledge. This highlights the idea that a skill cannot be considered to have been learned until retention over time is demonstrated [13]. Notably, the explicit group showed a more marked decrease than the implicit group after three months, which was in accordance with Poolton, Masters and Maxwell [8], suggesting that implicit learning may result in less decrease in performance, as the directly accumulated procedural knowledge is left to more effectively support motor performance. This difference in decrease in performance between groups was even found after six months of retention.

Another explanation could be that the explicit learners may have relied more on declarative knowledge, which suggests that retrieval from working memory may have been subject to decay, possibly due to some forgotten instructions. This differs from Maxwell, Masters and Eves [10], who found no differences between groups. However, their study conducted the retention test after just 72 h, at which time the participants probably still had declarative knowledge accessible in their working memory. Furthermore, cognitive processes continue to develop during adolescence, and individuals may vary in terms of their working memory capacity [17]. The volume of verbal instructions may also have been too high, potentially limiting the participants’ conscious attention to their own movement, given their average age of 10 years [18]. However, in the present study, no information about declarative knowledge and other cognitive processes related to the performance of the participants in the groups was gathered to support the explanation. Nevertheless, this information was not collected to prevent the implicit learners from starting to think about the task.

It seems that the implicit learning condition may have promoted a more effective learning experience over a longer time, as indicated by the lesser decrease in performance after six months for the implicit learners, as it could make the children unaware of errors rather than focusing on mistaken attempts; thus, shifting the focus away from performance-oriented environments, and minimising the impact of performance pressure [19]. Lola and Tzetzis [19] suggested that the dual-task paradigm provided confidence in the implicit learners regarding their ability to execute the specific task while reducing their meta-knowledge, with no rules to recall. As a result, it made the task ‘look’ easier, creating a concept of successful execution and a sense of pleasure. In this manner, implicit learning may align better with children’s preferred mode of learning through active play and trial-and-error strategies [20].

Both groups decreased their performance after three and six months, which was expected as learning a novel task without repetition over a long period would normally decrease performance over time due to a decrease in strength and coordination [21]. The findings of the present study highlight the importance of considering retention over an extended period to investigate learning a novel task after an extended hiatus, as different learning methods appear to result in different performance outcomes over extended periods without training. Furthermore, it would be interesting to see how both groups would develop their performance again with some repetition training, which should be included in future studies.

### Limitations

This study has some limitations as some participants had prior experience with the training task, potentially affecting the pure implicit learning group. Future studies should recruit participants without any previous experience. Another limitation is the homogeneity of the participant group in terms of age and amount of gymnastics training. Future studies should consider a wider range of ages and varying levels of initial skill to determine if these findings hold across different populations. Furthermore, independent judges were used who had years of experience in judging gymnastic performance and who were blinded to prevent detection bias. The judgement of performance between the two showed a significant difference, which is a possible indication of poor accuracy of measurement. However, it seemed to be a systematic bias as a high level of agreement and consistency was found, indicating a high reliability of the assessments.

## 5. Conclusions

Both learning conditions have effectively improved skill acquisition in gymnastic athletes after only four half-hour training sessions. However, the explicit group showed a more marked decrease than the implicit group during the retention tests (especially after 3 months), which was probably caused by the decay of their reliance on the retrieval of declarative knowledge from working memory. Implicit learning may result in a lower decrease in performance, as the directly accumulated procedural knowledge may deteriorate more slowly. The current findings highlight the importance of considering retention tests over an extended period to monitor performance development after learning a novel task through different learning strategies, as they can differ over time. Based upon the present findings, it seems that implicit learning of a novel task in gymnastics is more robust over time (6 months) without extra training than explicit learning of the same task, indicating the importance of implicit learning of motor tasks over time.

## 6. Practical Applications

Given the sustained performance observed in the implicit learning group after 6 months of retention compared to the explicit learning group, sports educators and coaches might consider integrating more implicit learning tasks into early training stages, particularly for skills where long-term retention is critical, like a front flip in gymnastics.

## Figures and Tables

**Figure 1 behavsci-14-00798-f001:**
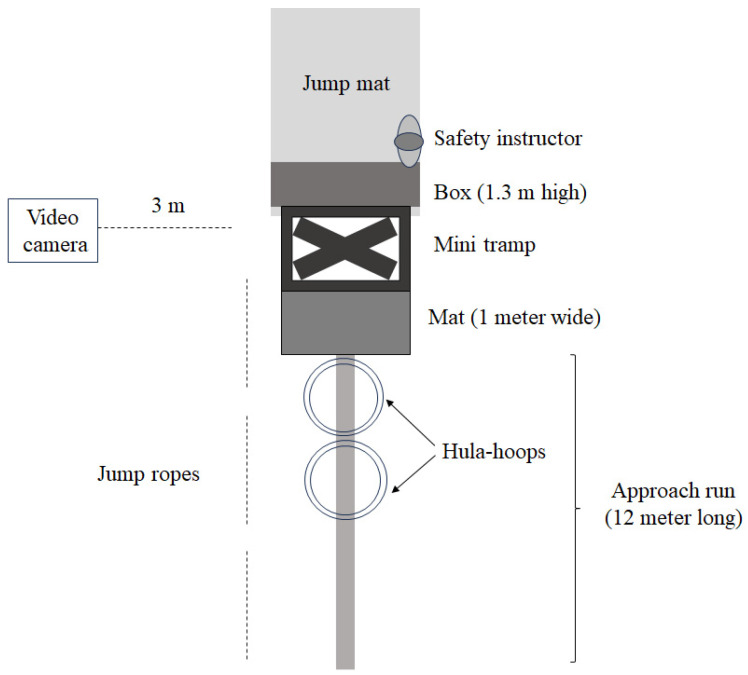
Illustration of training and test set-up.

**Figure 2 behavsci-14-00798-f002:**
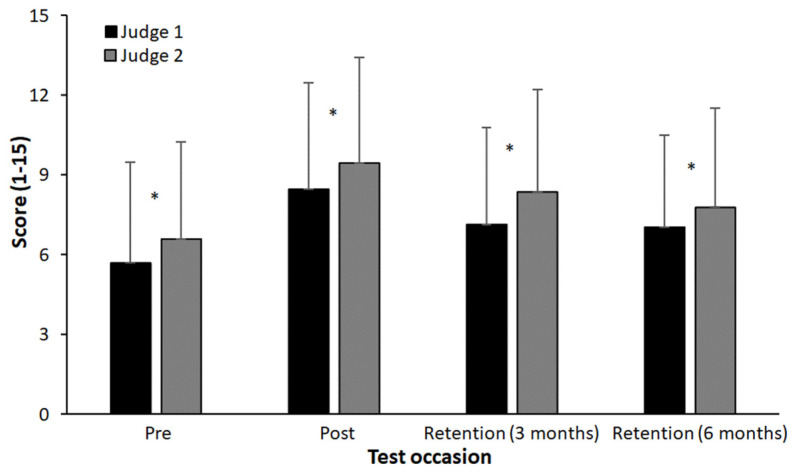
Mean (±SD) scores for the two judges at each test occasion. * indicates a significant difference between judges on a *p* < 0.05 level at this test occasion.

**Figure 3 behavsci-14-00798-f003:**
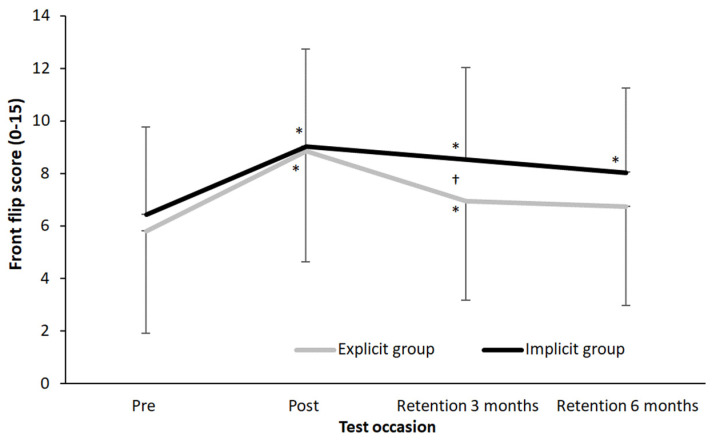
Average (±SD) front-flip performance per implicit and explicit training group on each test occasion. * indicates a significant difference with the previous test occasion on a *p* < 0.05 level. † indicates a significant difference in change from test to test between the two groups on a *p* < 0.05 level.

## Data Availability

The data presented in this study are available on request from the corresponding author. The data are not publicly available due to the national laws of the Norwegian government on privacy.

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
