# Peer review of "Comparison of Effects of Implicit versus Explicit Learning of a Novel Skill in Young Gymnastic Athletes"

_behavsci, 2024, doi:10.3390/bs14090798_

Round 1

Reviewer 1 Report

Comments and Suggestions for Authors

The article is focused on comparing the effects of explicit learning with implicit learning, using a dual-task paradigm to learn a novel skill. The topic is fascinating, and the findings are valuable to young athletes and their coaches. They can also be helpful in other scientific fields. The article is written well, but a few areas need to be revised. 

1. The article did not review much research on the topic. Adding a literature review will help readers understand the previous work on explicit and implicit learning in physical education and supports. It will also help readers understand the relationship established in the earlier research on global and local settings. 

2. Adding a table on participants' background information will help readers collect basic information quickly. 

3. The discussion section did not argue the main findings of the current study against the previous evidence. This is mainly because a literature review was not added, so not many studies were discussed to support our findings and different findings from prior research. 

4. The study implications are not provided in detail. A new section on the practical implications of this study's findings would be better. It would help the academic community better apply this study's conclusions in everyday life. 

5. The limitations in the discussion session are acceptable. However, I suggest that the conclusion section look much better in this study or that a separate section for limitations be created. 

Author Response

The article is focused on comparing the effects of explicit learning with implicit learning, using a dual-task paradigm to learn a novel skill. The topic is fascinating, and the findings are valuable to young athletes and their coaches. They can also be helpful in other scientific fields. The article is written well, but a few areas need to be revised.

  1. The article did not review much research on the topic. Adding a literature review will help readers understand the previous work on explicit and implicit learning in physical education and supports. It will also help readers understand the relationship established in the earlier research on global and local settings.

Response: We have explained from line 37-56 a short literature review. However, there are not many studies done with this setting with a dual task, which makes it difficult to come with a large literature review as not many studies have investigated this with this concept on implicit learning with a dual task. Furthermore, did we see that the most important part of the introduction was missing: the research question with the hypothesis. This is now included to the introduction and thereby we think that the introduction now is very clear for the reader as we think that it follows a clear line of story. We hope that the reviewer agrees with the changes.

  1. Adding a table on participants' background information will help readers collect basic information quickly.

Response: We think that we have already shown in the text what the background of the subjects is. Since they were only 8 years old they have not much background what also the point of the study was. The information about the background is shown in lines 62-64. We hope that the reviewer agrees that this is enough information.

  1. The discussion section did not argue the main findings of the current study against the previous evidence. This is mainly because a literature review was not added, so not many studies were discussed to support our findings and different findings from prior research.

Response: As mention on the comment in the introduction, not many studies were performed with this concept on implicit learning of a novel motor task with the dual task. We have tried to compare it with the earlier studies on this topic as much as possible. When the reviewer reads lines 185 to 235 he/she will see that we refer to earlier studies on this topic. We hope that the reviewer is informed enough about the findings in relation to earlier studies as we think that we have done this.

  1. The study implications are not provided in detail. A new section on the practical implications of this study's findings would be better. It would help the academic community better apply this study's conclusions in everyday life.

Response: We have added an extra section on practical applications to the text in which we give a practical application of the study for coaches and sports educators. We hope that the reviewer is satisfied with this.

  1. The limitations in the discussion session are acceptable. However, I suggest that the conclusion section look much better in this study or that a separate section for limitations be created.

Response: In the present manuscript the limitations are shown in an own paragraph, which we thought was already enough. We have now put a header above the paragraph to show the limitations in a own section.

Reviewer 2 Report

Comments and Suggestions for Authors

Dear authors,

In my opinion, there are no critical points regarding your study. The study has a clear design and all the variables collected are reported in a comprehensible manner. 

I have only formulated possible extensions and queries in the PDF.

Greetings

Author Response

Answers to the comments of the reviewer, which were send in a pdf file.

Line 90: We have changed this sentences to avoid confusion.

Line 92: stratified means that the participants were ranked after the pretest upon performance and then randomly assigned to one of the groups. So number 1 in explicit group, number 2 in implicit, 3 in explicit etc. This is to assure that performances between the groups are equal at pretest. This is a normal procedure in this type of experimental design. We have included this to the text:  (based upon rank at performance at pretest)

Line 111: We have written more information about the standardized feedback. The other group did not get any feedback, which is now also mentioned.

Line 114: yes just before they start running. We have included this to the text.

Line 122: We have tested the normality of data distribution and confirmed using the Shapiro-Wilk test.  This is now mentioned in the text.

Line 156: It would not be of interest to calculate the % change, because this does not count much in this set up as the measurements are based upon judge scores. Also the exact numbers would not be so interesting due to these judge scores, but the development over time as shown in the figure are much easier to follow in our opinion for the story. We hope that the reviewer agrees with our opinion that it is easier for the readers to follow by the shown figure.

Reviewer 3 Report

Comments and Suggestions for Authors

The paper lacks supporting literature reviews as they help to identify trends, patterns, and inconsistencies in the literature, providing a comprehensive understanding of the topic.

Authors also need to improve the detailed elaboration of the experimental steps within the study context; it is important to strengthen the positive points of the information. Detailed steps allow other researchers to replicate the study, which is a fundamental aspect of the scientific method. Replication helps verify the findings and establish their reliability and validity. The elaboration can also be represented by pictures of the activities.

Findings should also be improved; that is, a short or brief explanation of the findings, presented by the paper. Detailed findings provide a clear and comprehensive understanding of the research results, enabling readers to grasp the outcomes and their implications fully. Detailed findings facilitate a thorough analysis of the data. This includes identifying patterns, trends, and anomalies that might not be evident with a superficial overview.

Discussions are satisfactorily organized; however, practical and theoretical implications should be detailed and improved. Theoretical implications contribute to the advancement of knowledge by extending, refining, or challenging existing theories. Practical implications can inform policy development and reform by providing evidence-based insights that policymakers can use to create or modify policies.

Author Response

The paper lacks supporting literature reviews as they help to identify trends, patterns, and inconsistencies in the literature, providing a comprehensive understanding of the topic.

response: We have explained from line 37-56 a short literature review. However, there are not many studies done with this setting with a dual task, which makes it difficult to come with a large literature review as not many studies have investigated this with this concept on implicit learning with a dual task. Furthermore, did we see that the most important part of the introduction was missing: the research question with the hypothesis. This is now included to the introduction and thereby we think that the introduction now is very clear for the reader as we think that it follows a clear line of story. We hope that the reviewer agrees with the changes.

Authors also need to improve the detailed elaboration of the experimental steps within the study context; it is important to strengthen the positive points of the information. Detailed steps allow other researchers to replicate the study, which is a fundamental aspect of the scientific method. Replication helps verify the findings and establish their reliability and validity. The elaboration can also be represented by pictures of the activities.

Response: Due to the policy of the national rules we are not allowed to have pictures of the activities as the subjects are under 18 years old. So we don’t have pictures to show. Furthermore, have we tried to show how the set up was during training and testing by figure 1 and the text in lines 105-120.  

Findings should also be improved; that is, a short or brief explanation of the findings, presented by the paper. Detailed findings provide a clear and comprehensive understanding of the research results, enabling readers to grasp the outcomes and their implications fully. Detailed findings facilitate a thorough analysis of the data. This includes identifying patterns, trends, and anomalies that might not be evident with a superficial overview.

Response: We have tried to be detailed and concise with the findings. We can’t be more detailed than what we showed in the figures as we only had two groups with 4 measuring moments. In our opinion and also that of reviewer two are the results very clear. If the reviewer think that we have to be more detailed, could thew please specify what he/she means with that so we can conduct the evt. changes.

Discussions are satisfactorily organized; however, practical and theoretical implications should be detailed and improved. Theoretical implications contribute to the advancement of knowledge by extending, refining, or challenging existing theories. Practical implications can inform policy development and reform by providing evidence-based insights that policymakers can use to create or modify policies.

Response: We have added an extra sentence about the theoretical implication to the conclusion part and an extra section about the practical implications of the present study. We hope that the reviewer is satisfied with this.